**Data Availability Statement:** Data cannot be shared publicly because of the IRB's policy to store

# Impact of staffing model conversion from a mandatory critical care consultation model to a closed unit model in the medical intensive care unit

**Sung Jun Ko**[1], **Jaeyoung Cho**[2], **Sun Mi Choi**[2], **Young Sik Park**[2], **Chang-Hoon Lee**[2], **Chul-Gyu Yoo**[2], **Jinwoo Lee**[2]*, **Sang-Min Lee**[2]*

1 Department of Internal Medicine, Wonkwang University Sanbon Hospital, Gunpo, Republic of Korea,
2 Division of Pulmonary and Critical Care Medicine, Department of Internal Medicine, Seoul National University College of Medicine, Seoul National University Hospital, Seoul, Republic of Korea

* realrain7@gmail.com (JL); sangmin2@snu.ac.kr (SML)

## Abstract

### Background

The intensive care unit (ICU) staffing model affects clinical outcomes of critically ill patients. However, the benefits of a closed unit model have not been extensively compared to those of a mandatory critical care consultation model.

### Methods

This retrospective before-after study included patients admitted to the medical ICU. Anthropometric data, admission reason, Acute Physiology and Chronic Health Evaluation II score, Eastern Cooperative Oncology Group grade, survival status, length of stay (LOS) in the ICU, duration of mechanical ventilator care, and occurrence of ventilator-associated pneumonia (VAP) were recorded. The staffing model of the medical ICU was changed from a mandatory critical care consultation model to a closed unit model in September 2017, and indices before and after the conversion were compared.

### Results

A total of 1,526 patients were included in the analysis. The mean age was 64.5 years, and 954 (62.5%) patients were men. The mean LOS in the ICU among survivors was shorter in the closed unit model than in the mandatory critical care consultation model by multiple regression analysis (5.5 vs. 6.7 days; $p = 0.005$). Central venous catheter insertion (38.5% vs. 51.9%; $p < 0.001$) and VAP (3.5% vs. 8.6%; $p < 0.001$) were less frequent in the closed unit model group than in the mandatory critical care consultation model group. After adjusting for confounders, the closed unit model group had decreased ICU mortality (adjusted odds ratio 0.65; $p < 0.001$) and shortened LOS in the ICU compared to the mandatory critical care consultation model group.

the data in a password-protected file. Data are available from the Seoul National University Hospital IRB for researchers who meet the criteria for access to confidential data (SNUH IRB e-mail for contact: cris@bri.snuh.org).

**Funding:** The authors received no specific funding for this work.

**Competing interests:** The authors have declared that no competing interests exist.

## Conclusion

The closed unit model was superior to the mandatory critical care consultation model in terms of ICU mortality and LOS among ICU survivors.

## Introduction

The intensive care unit (ICU) is one of the most specialized units in hospitals. Critically ill patients have various comorbidities and need critical support, such as mechanical ventilation or renal replacement therapy (RRT), which requires skillful workmanship and extensive knowledge. Decisions in the ICU need to be accurate and prompt to respond to rapid changes in deteriorating patients. Hence, it is widely recommended that ICU physicians be experienced clinicians in critical care medicine.

Intensivists are board-certified experts in providing care for critically ill patients. As critical care medicine has become a distinct specialty, the need for specialized critical care physicians continues to grow worldwide. In South Korea, the subspecialty system for critical care medicine started in 2008 [1], and there were over 1,500 intensivists by 2019. However, there is an unmet need due to their high-demand and uneven distribution [2].

Although many institutions have ICUs, the staffing models of each ICU differ according to the number of available intensivists and the economic and cultural situations. Pronovost et al. classified ICU staffing models of intensivists into four groups: 1) a closed unit where the intensivist is the patient's primary attending physician, 2) a mandatory critical care consultation model where every patient admitted to the ICU receives a critical care consultation, 3) an elective critical care consultation model where the intensivist is involved only when needed, and 4) a model with no intensivist available. The former two groups were further classified as high-intensity staffing models, and the latter two groups were classified as low-intensity staffing models [3].

Many studies on the impact of the staffing model of intensivists in the ICU have routinely compared the high-intensity and low-intensity staffing models, revealing that a high-intensity staffing model was associated with reduced ICU and hospital mortality compared to a low-intensity model [4–14]. Several meta-analyses have shown similar results [3, 15, 16]. However, some studies have reported contradictory results [17, 18]. This subject has rarely been approached in Asian ICUs, and further studies are required to seek a plausible explanation for the differences in outcomes according to changes in the ICU staffing models. This study aimed to compare the closed unit model to the mandatory critical care consultation model and evaluate the superiority of the closed unit model in terms of outcomes in critically ill Asian patients.

## Methods

### 1. Study design and participants

This retrospective before-after cohort study included patients admitted to the medical ICU of a university-affiliated teaching hospital. Patients aged > 19 years and who were admitted to the medical ICU between January 2016 and August 2018 were included. Until August 2017, all patients admitted to the medical ICU received mandatory consultation by a board-certified intensivist. Two intensivists board certified in Internal Medicine and Pulmonology consulted and supervised all patients in the medical ICU daily, but the original primary attending physician continued to be primary charge. From September 2017, the ICU staffing model was

changed from a mandatory critical care consultation model to a closed unit model. Patient care in the medical ICU was formally transferred to an intensivist. The dedicated intensivist was present in the ICU during the weekday daytime and was responsible for all patient care, including admission, management, and discharge. The same two intensivists were involved in patient management in both models. We compared the indices before and after the conversion to evaluate the advantages of the closed unit model and compared it to the mandatory critical care consultation model.

This study was approved by the Institutional Review Board (IRB) of Seoul National University Hospital (IRB No.1807-140-961). The requirement for informed consent was waived because of the retrospective design of this study.

## 2. Data collection

The following variables were recorded after reviewing the medical records: age, sex, Acute Physiology and Chronic Health Evaluation (APACHE) II score, Eastern Cooperative Oncology Group (ECOG) performance status grade, primary reason for ICU admission, cardiopulmonary resuscitation (CPR), referral to palliative care within 24 hours of ICU admission, treatments during the ICU stay, occurrence of ventilator-associated pneumonia (VAP) and delirium during ICU stay, ICU readmission (readmission within 48 hours of ICU discharge), survival status, length of stay (LOS) in the ICU and duration of mechanical ventilator (MV) care.

## 3. Statistical analysis

Participants were divided into two groups according to the staffing model and the baseline characteristics of the groups were compared. To determine the independent effect of staffing model on ICU mortality, we performed multiple logistic regression analysis by adjusting for age, sex, APACHE II score, ECOG grade, and reasons for ICU admission. The independent impact of the staffing model on the LOS in the ICU and duration of MV care were evaluated using multiple regression analysis.

Subgroup analyses were conducted according to the five reasons for admission: respiratory diagnosis, cardiovascular diagnosis, acute kidney injury, sepsis, and neurologic diagnosis. ICU mortality, LOS in the ICU, and duration of MV care were compared.

Statistical analyses were performed using SPSS software (version 25.0 for Windows; IBM SPSS Inc., Armonk, NY, USA) and R (version 4.0.0, https://www.R-project.org). All statistical tests were two-sided, and differences were considered statistically significant at $p < 0.05$.

## Results

### 1. Study population and treatment in the ICU

A total of 1,657 patients (1,076 and 581 patients in the mandatory critical consultation model group and the closed unit model group, respecively) were admitted to the medical ICU between January 2016 and August 2018. Among them, 131 patients (89 [8.3%] and 42 [7.2%] patients in mandatory critical care consultation model group and the closed unit model group, respectively) were excluded due to incomplete medical records; hence, 1,526 patients were included in the final analysis. The patients were categorized into two groups—the mandatory critical care consultation model group (987 [64.7%] patients) and the closed unit model group (539 [35.3%] patients).

The baseline characteristics of the patients are presented in Table 1. The mean age was 64.5 years, and 954 (62.5%) patients were men. Patients in the closed unit model group had higher

**Table 1. Baseline characteristics of the participants.**

| Characteristic | Mandatory critical care consultation model group (n = 987) | Closed unit model group (n = 539) | p value |
|---|---|---|---|
| Age, years | 64.5 ± 14.8 | 64.9 ± 15.1 | 0.592 |
| Male sex (n, %) | 616 (62.4%) | 338 (62.7%) | 0.909 |
| APACHE II score | 21.3 ± 9.8 | 22.7 ± 9.5 | 0.008 |
| ECOG grade | 3.0 ± 1.1 | 3.3 ± 0.9 | <0.001 |
| ICU admission diagnosis† | | | |
| Respiratory, n (%) | 642 (65.0%) | 378 (70.1%) | 0.044 |
| Cardiovascular, n (%) | 191 (19.4%) | 92 (17.1%) | 0.273 |
| Acute kidney injury, n (%) | 163 (16.5%) | 106 (19.7%) | 0.123 |
| Sepsis, n (%) | 152 (15.4%) | 89 (16.5%) | 0.569 |
| Neurologic, n (%) | 47 (4.8%) | 23 (4.3%) | 0.659 |
| Others††, n (%) | 123 (12.5%) | 74 (13.7%) | 0.480 |
| CPR within 24 hours of ICU admission, n (%) | 128 / 971 (13.2%) | 73 / 528 (13.8%) | 0.727 |
| Referred to palliative care within 24 hours of ICU admission, n (%) | 84 / 962 (8.7%) | 51 / 528 (9.7%) | 0.551 |

\* Values are presented as number/total number (%) for categorical variables or mean ± standard deviation for continuous variables.

\*APACHE II score, Acute Physiology and Chronic Health Evaluation II score; ECOG, Eastern Cooperative Oncology Group; ICU, intensive care unit; CPR, cardiopulmonary resuscitation.

† Multiple choices were available for ICU admission diagnosis.

†† Others included gastrointestinal bleeding, for close observation after surgery or procedure, psychiatric, poisoning and etc.

APACHE II scores (22.7 vs. 21.3; $p$ = 0.008), ECOG grade (3.3 vs. 3.0; $p$ < 0.001) and higher proportion of patients with ECOG grade ≥3 (84.8% vs. 73.0%; $p$ < 0.001) than those in the mandatory critical care consultation model group. Respiratory failure was the most common reason for ICU admission in both groups, but it was more frequent in the closed unit model group than in the mandatory critical care consultation model group (70.1% vs. 65.0%; $p$ = 0.044).

During the ICU stay, 1,033 (67.7%) patients were mechanically ventilated, and 515 (33.7%) patients died in the ICU. The use of central venous catheters was less frequent in the closed unit model group than in the mandatory critical care consultation model group (38.5% vs. 51.9%; $p$ < 0.001), but the rates of other treatment options including RRT, tracheostomy, and extracorporeal membrane oxygenation (ECMO) did not differ between the staffing models. The occurrence of VAP was significantly lower in the closed unit model group than in the mandatory critical care consultation model group (3.5% vs. 8.6%; $p$ < 0.001). The difference in ICU mortality was not statistically significant between the staffing models (31.2% vs. 35.2%; $p$ = 0.115), but the overall LOS in the ICU was shorter in the closed unit model group than in the mandatory critical care consultation model group (6.4 vs. 7.3 days; $p$ = 0.024). The rates of ICU readmission did not differ between the staffing models (0.7% vs. 1.5%; $p$ = 0.190) (Table 2).

## 2. Factors associated with all-cause ICU mortality

After multiple logistic regression adjusted for age, sex, APACHE II score, and ECOG grade, conversion to the closed unit model decreased ICU mortality by 35% ($p$ < 0.001). A high APACHE II score, ECOG grade, and ICU admission for acute kidney injury or sepsis were independent risk factors for ICU mortality (Table 3).

**Table 2. The study participants' treatments and clinical outcomes.**

| Treatment and outcome | Mandatory critical care consultation model group (n = 987) | Closed unit model group (n = 539) | p value |
|---|---|---|---|
| Invasive procedures | | | |
| Mechanical ventilation, n (%) | 652 / 987 (66.1%) | 381 / 539 (70.7%) | 0.065 |
| RRT, n (%) | 313 / 948 (33.0%) | 153 / 514 (29.8%) | 0.203 |
| Tracheostomy, n (%) | 150 / 945 (15.9%) | 77 / 512 (15.0%) | 0.675 |
| ECMO, n (%) | 36 / 950 (3.8%) | 27 / 520 (5.2%) | 0.201 |
| Central venous catheter, n (%) | 498 / 959 (51.9%) | 198 / 514 (38.5%) | <0.001 |
| VAP, n (%) | 81 / 939 (8.6%) | 18 / 517 (3.5%) | <0.001 |
| Delirium, n (%) | 104 / 937 (11.1%) | 50 / 516 (9.7%) | 0.404 |
| In-ICU mortality, n (%) | 347 (35.2%) | 168 (31.2%) | 0.115 |
| Overall LOS in ICU, days | 7.3 ± 8.1 | 6.4 ± 7.6 | 0.024 |
| LOS in ICU among survivors | 6.7 ± 6.9 | 5.5 ± 6.0 | 0.005 |
| LOS in ICU among non-survivors | 8.5 ± 9.8 | 8.4 ± 10.0 | 0.841 |
| The overall duration of MV care, days | 2.5 ± 4.6 | 2.2 ± 4.2 | 0.326 |
| MV care duration among survivors | 2.5 ± 4.5 | 2.2 ± 4.1 | 0.439 |
| MV care duration among non-survivors | 2.6 ± 4.9 | 2.1 ± 4.4 | 0.537 |
| ICU readmission rate, n (%) | 15 (1.5%) | 4 (0.7%) | 0.190 |

*Values are presented as number/total number (%) for categorical variables or mean ± standard deviation for continuous variables.

*RRT, renal replacement therapy; ECMO, extracorporeal membrane oxygenation; VAP, ventilator-associated pneumonia; ICU, intensive care unit; LOS, length of stay; MV, mechanical ventilator.

### 3. Factors associated with LOS in the ICU among survivors

We analyzed the association between the staffing model and LOS in the ICU among the 1,011 ICU survivors. After adjusting for age, sex, APACHE II score, and ECOG grade, patients in the closed unit model group had shorter LOS in the ICU by 1.88 days than those in the mandatory critical care consultation model group ($p < 0.001$). Patients admitted for respiratory failure or sepsis stayed longer in the ICU than other patients ($p < 0.001$ and $p = 0.022$, respectively) (Table 4).

### 4. Differences in outcomes according to the ICU admission diagnosis

The benefits of the closed unit model were more prominent among patients admitted to the ICU for respiratory or cardiovascular diseases. Among the patients with ICU admission diagnosis of respiratory failure, ICU mortality was lower (30.4% vs. 37.2%; $p = 0.028$), and the LOS in the ICU was shorter (6.3 ± 6.6 vs. 7.5 ± 7.1; $p = 0.027$) in the closed unit model group than in the mandatory critical care consultation model group. Similar findings were observed in patients admitted for cardiovascular failure. The LOS in the ICU and the duration of MV care were shorter (4.4 ± 5.4 vs. 7.4 ± 7.6; $p = 0.01$ and 1.1 ± 3.1 vs. 3.0 ± 4.0; $p = 0.002$, respectively) in the closed unit model group than in the mandatory critical care consultation model group. A statistically significant difference in outcome was not observed in patients with an admission diagnosis of acute kidney injury, sepsis, or neurological diseases (Table 5).

## Discussion

Our study revealed that the closed unit model decreased ICU mortality and shortened LOS of critically ill patients compared to the mandatory critical care consultation model. Although many studies have suggested the superiority of the high-intensity staffing models over the low-intensity models, most studies have been conducted in the United States [4–9, 17, 18]. Only a

**Table 3. Independent predictors of in-ICU mortality by multiple logistic regression analysis.**

| Independent variables | Univariate analysis | | Multivariate analysis | |
|---|---|---|---|---|
| | Odds ratio | *p* value | Odds ratio | *p* value |
| | (95% CI) | | (95% CI) | |
| Age | 0.99 (0.99–1.00) | 0.059 | 0.99 (0.98–0.99) | 0.001 |
| Sex | 1.06 (0.85–1.32) | 0.579 | | |
| APACHE II score | 1.08 (1.06–1.09) | <0.001 | 1.07 (1.05–1.08) | <0.001 |
| (increase in 1 point) | | | | |
| ECOG grade | 1.55 (1.38–1.74) | <0.001 | 1.37 (1.21–1.56) | <0.001 |
| (increase in 1 point) | | | | |
| Closed unit model | 0.84 (0.67–1.04) | 0.115 | 0.65 (0.51–0.83) | <0.001 |
| ICU admission diagnosis | | | | |
| Respiratory | 1.14 (0.91–1.43) | 0.262 | | |
| Cardiovascular | 1.18 (0.90–1.54) | 0.237 | | |
| Acute kidney injury | 2.24 (1.71–2.93) | <0.001 | 1.67 (1.25–2.23) | <0.001 |
| Sepsis | 2.74 (2.07–3.63) | <0.001 | 1.83 (1.34–2.49) | <0.001 |
| Neurologic | 1.33 (0.80–2.16) | 0.259 | | |

*ICU, intensive care unit; APACHE II score, Acute Physiology and Chronic Health Evaluation II score; ECOG, Eastern Cooperative Oncology Group.

few studies have been conducted in Asia, but with a limited number of surgical ICU patients [10] or postoperative patients [11]. A Turkish study showed an improved survival rate after conversion to a closed unit model, but the sample size was relatively small (<40% of this study population) and a detailed description of the participants' characteristics was not provided [12]. A retrospective Japanese study reported better survival in patients with sepsis in the closed unit model than in the open unit model among 35 heterogeneous ICUs, but detailed description of each ICU closed unit was not available and the results could not be applied to the general ICU population [13].

One of the meaningful findings of our study is that the closed unit model was associated with better outcomes than the mandatory critical care consultation model involving the same intensivists. Most of the studies reporting the beneficial effect of a closed unit model compared

**Table 4. Independent predictors of ICU length of stay among survivors by multiple regression analysis.**

| Independent variables | Univariate analysis | | Multivariate analysis | |
|---|---|---|---|---|
| | β-coefficient | *p* value | β-coefficient | *p* value |
| Age | -0.03 | 0.022 | -0.06 | <0.001 |
| Sex | 0.42 | 0.322 | | |
| APACHE II score | 0.16 | <0.001 | 0.16 | <0.001 |
| ECOG grade | 0.78 | <0.001 | 0.56 | 0.004 |
| Closed unit model | -1.20 | 0.005 | -1.88 | <0.001 |
| ICU admission diagnosis | | | | |
| Respiratory | 2.24 | <0.001 | 2.35 | <0.001 |
| Cardiovascular | 0.30 | 0.579 | | |
| Acute kidney injury | 0.31 | 0.604 | | |
| Sepsis | 1.40 | 0.034 | 1.46 | 0.022 |
| Neurologic | 1.68 | 0.106 | | |

*ICU: Intensive Care Unit, APACHE II score: Acute Physiology and Chronic Health Evaluation II score, RRT: Renal Replacement Therapy.

**Table 5. Patient outcomes changes in the five subgroups according to the staffing model of ICU.**

| Patient outcome | Mandatory critical care consultation model | Closed unit model | *p* value |
|---|---|---|---|
| Overall patients | n = 987 | n = 539 | |
| Mortality, n (%) | 347 (35.2%) | 168 (31.2%) | 0.115 |
| LOS in ICU of survivors, days | 6.7 ± 6.9 | 5.5 ± 6.0 | 0.005 |
| Duration of MV care of survivors, days | 2.5 ± 4.5 | 2.2 ± 4.1 | 0.439 |
| Patients with admission diagnosis of respiratory disease | n = 642 | n = 378 | |
| Mortality, n (%) | 239 (37.2%) | 115 (30.4%) | 0.028 |
| LOS in ICU of survivors, days | 7.5 ± 7.1 | 6.3 ± 6.6 | 0.027 |
| Duration of MV care of survivors, days | 3.5 ± 5.2 | 2.9 ± 4.7 | 0.187 |
| Patients with admission diagnosis of cardiovascular disease | n = 191 | n = 92 | |
| Mortality, n (%) | 67 (35.1%) | 37 (40.2%) | 0.401 |
| LOS in ICU of survivors, days | 7.4 ± 7.6 | 4.4 ± 5.4 | 0.010 |
| Duration of MV care of survivors, days | 3.0 ± 4.0 | 1.1 ± 3.1 | 0.002 |
| Patients with admission diagnosis of acute kidney injury | n = 163 | n = 106 | |
| Mortality, n (%) | 84 (51.5%) | 49 (46.2%) | 0.395 |
| LOS in ICU of survivors, days | 6.3 ± 4.7 | 6.8 ± 6.8 | 0.549 |
| Duration of MV care of survivors, days | 1.7 ± 5.2 | 2.6 ± 5.1 | 0.333 |
| Patients with admission diagnosis of sepsis | n = 152 | n = 89 | |
| Mortality, n (%) | 84 (55.3%) | 46 (51.7%) | 0.591 |
| LOS in ICU of survivors, days | 7.7 ± 6.9 | 7.1 ± 7.1 | 0.644 |
| Duration of MV care of survivors, days | 2.1 ± 3.7 | 2.4 ± 3.7 | 0.753 |
| Patients with admission diagnosis of neurologic disease | n = 47 | n = 23 | |
| Mortality, n (%) | 17 (36.2%) | 11 (47.8%) | 0.350 |
| LOS in ICU of survivors, days | 9.3 ± 9.2 | 4.3 ± 3.2 | 0.074 |
| Duration of MV care of survivors, days | 3.5 ± 9.2 | 1.0 ± 2.0 | 0.405 |

*Values are presented as number (%) for categorical variables or mean ± standard deviation for continuous variables.

*ICU, intensive care unit; LOS, length of stay; MV, mechanical ventilator.

it to an open unit model. Further, they did not indicate whether the open unit model was a mandatory or elective critical care consultation model, or a no intensivist model [4, 5, 10–12, 14]. It was also unclear whether there were changes in the intensivists involved in patient care. This is also a limitation of the studies that include multiple institutes [7–9, 13, 17, 18].

Although the patients had higher APACHE II scores at admission, the closed unit model led to improved outcomes in critically ill patients. Although speculative, the active admission triage of the intensivists in the closed unit model might have resulted in the admission of patients with higher severity. The improved ICU outcomes in the closed unit model might be due to the decrease in ICU care complications. Although the invasive treatment/procedures performed in both groups were not different, the frequency of central venous catheter insertion was significantly lower in the closed unit model group than in the mandatory critical care consultation model group. This may have led to a lower rate of ICU-acquired infections. This assumption was further supported by the significantly lower VAP, another frequent ICU-acquired infection, in the closed unit model group than in the mandatory critical care consultation model group. Other studies have also shown a reduced VAP rate after conversion to a closed unit model [19, 20].

Subgroup analysis in this study showed that the benefits of the closed unit model were mostly found in patients with respiratory or cardiovascular failure. Timely application and handling of equipments by specialists, including mechanical ventilators, noninvasive

ventilators, and ECMO, may be associated with the beneficial results in these subgroups [21–24]. Early access to diagnostic tools such as bronchoscopy, echocardiography, and ultrasonography may also have played a significant role [25–28].

Our study has several limitations. First, due to the before-after observational design of this study, it was possible that factors other than the staffing model may have affected the outcome. Further, the outcome changes might have been due to advances in medicine rather than changes in staffing models [29]. However, the same two intensivists participated in the treatment of enrolled patients in both models. In addition, the proportion of patients referred to palliative care within 24 hours of ICU admission, a possible surrogate for inappropriate admission, was not different between the groups. Second, caution is needed when interpreting the results of a single-center study. ICU conditions differ greatly from one nation to another and also within one nation. However, this study demonstrated that changes in the staffing model can improve outcomes in the medical ICU of an Asian country and identified subgroups that might benefit most from the changes. Third, the number of patients in the mandatory consultation model group was twice as high as that than in the closed unit model group. This was due to differences in time period before and after changing the staffing model. Although equal-sized groups have maximal statistical power, we believe that the smaller size of the group in our study (n = 539) was large enough to detect clinically significant differences between the two groups.

In conclusion, the closed unit model proved to be superior to the mandatory critical care consultation model in terms of ICU mortality and LOS in the ICU. The beneficial effects of the closed unit model were more prominent in patients admitted for respiratory and cardiovascular failure.

## Author Contributions

**Conceptualization:** Sung Jun Ko, Jinwoo Lee, Sang-Min Lee.

**Data curation:** Sung Jun Ko.

**Formal analysis:** Sung Jun Ko.

**Investigation:** Sung Jun Ko, Jaeyoung Cho, Sun Mi Choi, Young Sik Park, Chang-Hoon Lee, Chul-Gyu Yoo, Jinwoo Lee, Sang-Min Lee.

**Methodology:** Sung Jun Ko, Jaeyoung Cho, Sun Mi Choi, Young Sik Park, Chang-Hoon Lee, Chul-Gyu Yoo, Jinwoo Lee, Sang-Min Lee.

**Supervision:** Jinwoo Lee, Sang-Min Lee.

**Writing – original draft:** Sung Jun Ko.

**Writing – review & editing:** Jinwoo Lee, Sang-Min Lee.

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
