## [Decision Letter · Decision Letter 0]

7 Jul 2021

PONE-D-21-19587

Impact of staffing model conversion from a mandatory critical care consultation model to a closed unit model in the medical intensive care unit

PLOS ONE

Dear Dr. Lee,

Thank you for submitting your manuscript to PLOS ONE. After careful consideration, we feel that it has merit but does not fully meet PLOS ONE’s publication criteria as it currently stands. Therefore, we invite you to submit a revised version of the manuscript that addresses the points raised during the review process.

Please address the issues and revise accordingly.

We look forward to receiving your revised manuscript.

Kind regards,

Academic Editor

PLOS ONE

Journal Requirements:

Reviewers' comments:

Reviewer's Responses to Questions

**Comments to the Author**

1. Is the manuscript technically sound, and do the data support the conclusions?

Reviewer #1: Yes

Reviewer #2: Yes

Reviewer #3: Yes

2. Has the statistical analysis been performed appropriately and rigorously? 

Reviewer #1: I Don't Know

Reviewer #2: Yes

Reviewer #3: Yes

3. Have the authors made all data underlying the findings in their manuscript fully available?

Reviewer #1: No

Reviewer #2: Yes

Reviewer #3: Yes

4. Is the manuscript presented in an intelligible fashion and written in standard English?

Reviewer #1: Yes

Reviewer #2: Yes

Reviewer #3: Yes

5. Review Comments to the Author

Reviewer #1: Lee and colleagues present a restrospective analysis of different clinical outcomes (mainly mortality, length of stay (LOS), rates of central venous catheterization and ventilator associated pneumonia (VAP)) in patients treated in a medical ICU in either a closed unit model (intensivist is the primary physician in charge) or a mandatory critical care consultation model (primary attending physician is in charge + mandatory consultations of an intensivist). Main findings a reduced LOS, central venous catheterization- and VAP-rate, and - in the multivariate analysis - a reduced mortality in the closed unit model compared with the mandatory critical care consultation model. Of course, this study carries with it the innate limitations of retrospective studies, which the authors acknowledge. The manuscript is overall well-written, some minor errors should be corrected.

Reviewer #2: I read with great interest this manuscript on the impact of the ICU staffing model conversion. This is a highly relevant article in the critical care world and I am in agreement with the authors on superiority of the closed unit model and on the unmet need for board-certified intensivists.

I felt the manuscript contained sound statistical analysis and was well-written and succinct in communication of pertinent information. Most of the questions/suggestions that I jotted down during review of the paper were answered and addressed in subsequent sections of the manuscript. Examples include definitions of four staffing models, discussion of limitations, subgroup analysis and characteristics, predictors of mortality, etc.

My final comments are outlined below, but my recommendation is to accept this manuscript for publication with minor revisions. I commend the authors on their rigorous work in this important study.

1. I agree that that ICU characteristics are highly variable among different sites within the same country and internationally. As such, intensivist training and board certification processes differ as well. Since the authors report best outcomes for patients with respiratory and cardiovascular conditions, would you comment on any additional training background of the intensivists in this study? In the US, the pathway to becoming an intensivist is available through critical care fellowship and board certification to anesthesiologists, surgeons, emergency physicians and internal medicine physicians, with the latter group often also co-trained in pulmonology. In other words, these different subgroups of intensivists bring additional skill sets to the ICU.

2. Was the comparison data on rate of ICU re-admissions available between the two staffing models? (i.e. % of patients downgraded from the ICU who were re-admitted to ICU during the same hospital stay). It would be an interesting outcome to track.

3. The “before” and “after” staffing model group sizes are at almost 2:1 ratio (1.8, to be exact). While there were adjustments in statistical analysis, this disparity would be worthwhile to include in the limitations section and discuss its potential effects on results.

4. Discussion section, Limitations, 2nd to last paragraph: “Factors other than staffing model may have affected the outcomes”. Can you provide some examples, either from the current study or from pertinent literature ? Any confounding variables?

5. Discussion section, 1st sentence: “shortened LOS of critically ill patients than the mandatory critical care consultation model”—“than” should be changed to “compared to”.

Reviewer #3: Dear editor, here you receive my review regarding the manuscript entitled ” Impact of staffing model conversion from a mandatory critical care consultation model to a closed unit model in the medical intensive care unit ”, with number PONE-D-21-19587.

The authors describe and present the results of a retrospective before and after cohort study, with patients included who were admitted to the medical ICU in a university teaching hospital. Between January 2016 and August 2017 patients received care on a mandatory consultation basis. From September 2017 till August 2018 patients were received medical care from a closed format unit model. Various indices were compared. The article is well written and easy to understand.

The need for personal informed consent from patients was waived by the medical ethical committee due to the retrospective design of the study.

I have a few comments to make.

There are many improvements made. n abstract and discussion important results are presented, i.e. CVL associated blood stream infections, catheter associated urinary tract infections and ventilator associated pneumonia and the influence on LOS and MV duration.

References are ok.

Page 1 the abstract: Here the order in presentation of either models is confusing, i.e., in the 3rd sentence of results first closed unit model (CUM) and second the mandatory critical care consultation model ( MCCCM) is mentioned. However the order of presentation is changed in the following sentence hen results of CVC and VAP are presented with opponent results. Please choose an order in presentation in the whole article? This will improve reading and understanding what is different. For instance, do I correctly understand that less CVC were used and there were more VAP’s in MCCCM?

The observation periods differ between the 2 models with 7 months more in the mandatory critical care consultation model. Please explain and discuss? This resulted in a different number of patients between groups, i.e. 987 vs 539 patients.

How many patients were not include in these 2 periods? In other words , what was the ratio or sample size of this cohort presented/studied in comparison with the total number of patients who were admitted tot ICU?

What was exactly the assumption or hypothesis before starting the study?

Ad Introduction, Line13 please write Pronovost instead of Pronovist

Ad Methods: L9 …during day and was responsible…? Is here missing maybe “during day and night, and was…” Do you mean 7 x 24 hours closed format/dedication, and weekend included?

Were there any differences regarding the time of discharge, i.e. outside time schedule 8:00-18:00 hours, with a possible association in (repeated) admission within 48 hours after discharge?

Ad statistical analysis: L7… to the five reasons… Here please describe in short what is meant? For instance: In short, ……..are the 5 organ related failures plus others?

Ad Results L7: patients in CUM had higher APPACHE II scores. Please explain? Could it be possible that with more dedication of the intensivists present there was more time to score the APACHE better, which could have led to the this difference?

Ad Table 1 ECOG-scale is presented as significantly different between groups, but with a mean of 3.0 vs 3.3. Statistically different, but I doubt whether of any clinical importance. Is it possible to divide between patients with ECOG≤2 and ≥3 for either group?

Table 1 Others with a remark, with stated “etc” What is the quantity of etc and what is exactly meant by etc? In total it is >12.5%

So LOS and MV duration are in favor of MCCCM. This is important for your ICU and conform previous study results. Whereas, when looking at patients with neurologic disorders (Table 5) the total number of patients may have not been enough to reach significance (p=0.074) in favor of MCCCM. Please explain and describe?

6. PLOS authors have the option to publish the peer review history of their article (what does this mean?). If published, this will include your full peer review and any attached files.

Reviewer #1: No

Reviewer #2: No

Reviewer #3: **Yes: **P. Bruins

---

## [Author Response · Author response to Decision Letter 0]

15 Aug 2021

Dear Dr. Emily Chenette,

Editor-in-Chief of PLoS One

We wish to thank the editor and reviewers for their response to our work and their constructive comments. We have addressed the concerns that were raised through further editing of the manuscript and believe it has significantly improved as a consequence of this process. We have highlighted all changes in the revised manuscript and have addressed all referees’ comments in the point-by-point responses below. We greatly appreciate the opportunity to submit this revised review manuscript and hope it is suitable for publication in PLoS One.

Reviewer #1: 

Lee and colleagues present a retrospective analysis of different clinical outcomes (mainly mortality, length of stay (LOS), rates of central venous catheterization and ventilator associated pneumonia (VAP) in patients treated in a medical ICU in either a closed unit model (intensivist is the primary physician in charge) or a mandatory critical care consultation model (primary attending physician is in charge + mandatory consultations of an intensivist). Main findings a reduced LOS, central venous catheterization- and VAP-rate, and - in the multivariate analysis - a reduced mortality in the closed unit model compared with the mandatory critical care consultation model. Of course, this study carries with it the innate limitations of retrospective studies, which the authors acknowledge. The manuscript is overall well-written, some minor errors should be corrected.

We thank the reviewer for this valuable comment. As suggested, we have gone through the revised article and corrected minor errors.

Introduction 13th line: Corrected “Pronovist” to “Pronovost”

Discussion 2nd line: Corrected “than” to “compared to”

Method: Study design and participants 10th line: Corrected “during day” to “during the weekday daytime”

Reviewer #2: 

I read with great interest this manuscript on the impact of the ICU staffing model conversion. This is a highly relevant article in the critical care world and I am in agreement with the authors on superiority of the closed unit model and on the unmet need for board-certified intensivists.

I felt the manuscript contained sound statistical analysis and was well-written and succinct in communication of pertinent information. Most of the questions/suggestions that I jotted down during review of the paper were answered and addressed in subsequent sections of the manuscript. Examples include definitions of four staffing models, discussion of limitations, subgroup analysis and characteristics, predictors of mortality, etc.

My final comments are outlined below, but my recommendation is to accept this manuscript for publication with minor revisions. I commend the authors on their rigorous work in this important study.

1. I agree that that ICU characteristics are highly variable among different sites within the same country and internationally. As such, intensivist training and board certification processes differ as well. Since the authors report best outcomes for patients with respiratory and cardiovascular conditions, would you comment on any additional training background of the intensivists in this study? In the US, the pathway to becoming an intensivist is available through critical care fellowship and board certification to anesthesiologists, surgeons, emergency physicians and internal medicine physicians, with the latter group often also co-trained in pulmonology. In other words, these different subgroups of intensivists bring additional skill sets to the ICU.

We thank the reviewer for this insightful comment. The pathway to becoming an intensivist is similar in South Korea. Anesthesiologists, surgeons, emergency physicians, neurologists, and internal medicine physicians can become an intensivist after adequate training through critical care fellowship. Most internal medicine physicians receive training in pulmonology before critical care fellowship. Both intensivists in this study were board certified in Internal Medicine and Pulmonology and we added this information in 10th line of Methods: Study design and participants.

2. Was the comparison data on rate of ICU re-admissions available between the two staffing models? (i.e. % of patients downgraded from the ICU who were re-admitted to ICU during the same hospital stay). It would be an interesting outcome to track.

We agree with the reviewer that ICU readmission is one of the important outcomes. We assessed rate of readmission (defined as readmission within 48 hours of ICU discharge), and found that although the rate of readmission was lower in the closed unit model (0.7% vs. 1.5%; p = 0.190), the number was too small to show any statistical significance. We added this result to the manuscript: 26-27 lines of Results: Study population and treatment in the ICU and Table 2.

3. The “before” and “after” staffing model group sizes are at almost 2:1 ratio (1.8, to be exact). While there were adjustments in statistical analysis, this disparity would be worthwhile to include in the limitations section and discuss its potential effects on results.

We thank the reviewer for this valuable comment. The duration of “before” staffing model was about several months longer than the “after” staffing model. This difference was the result of the timing of our study enrollment. We planned this study as soon as we believed we have gathered enough data to compare the different staffing models. 

We understand the reviewer’s concern and added this as a limitation of this study (50-54th lines of Discussion). 

4. Discussion section, Limitations, 2nd to last paragraph: “Factors other than staffing model may have affected the outcomes”. Can you provide some examples, either from the current study or from pertinent literature? Any confounding variables?

We thank the reviewer for this valuable comment. Although there were no confounding variables found during the retrospective analysis, the before-after observational design of this study in itself has its limitations. We cannot completely exclude the possibility that outcome improvements may be from advances in medicine (Curr Anesthesiol Rep 2013;3:65-72) rather than changes in staffing models. For example, decreased use of central venous catheter in this study may be the result of studies that have shown usual care is not inferior to early, goal-directed therapy in septic shock. (N Engl J Med 2017;376:2223-2234). We have added this explanation in the discussion section (41-42th lines of Discussion). 

5. Discussion section, 1st sentence: “shortened LOS of critically ill patients than the mandatory critical care consultation model”—“than” should be changed to “compared to”.

Thank you for correcting our mistake. Changes were made in 2nd line of Discussion as suggested.

Reviewer #3: 

Dear editor, here you receive my review regarding the manuscript entitled ” Impact of staffing model conversion from a mandatory critical care consultation model to a closed unit model in the medical intensive care unit ”, with number PONE-D-21-19587.

The authors describe and present the results of a retrospective before and after cohort study, with patients included who were admitted to the medical ICU in a university teaching hospital. Between January 2016 and August 2017 patients received care on a mandatory consultation basis. From September 2017 till August 2018 patients were received medical care from a closed format unit model. Various indices were compared. The article is well written and easy to understand.

The need for personal informed consent from patients was waived by the medical ethical committee due to the retrospective design of the study.

I have a few comments to make.

There are many improvements made. n abstract and discussion important results are presented, i.e. CVL associated blood stream infections, catheter associated urinary tract infections and ventilator associated pneumonia and the influence on LOS and MV duration.

References are ok.

* Page 1 the abstract: Here the order in presentation of either models is confusing, i.e., in the 3rd sentence of results first closed unit model (CUM) and second the mandatory critical care consultation model (MCCCM) is mentioned. However the order of presentation is changed in the following sentence hen results of CVC and VAP are presented with opponent results. Please choose an order in presentation in the whole article? This will improve reading and understanding what is different. For instance, do I correctly understand that less CVC were used and there were more VAP’s in MCCCM?

We apologize for the confusion. Both central venous catheter insertion and VAP were less frequent in the closed unit model (CUM). We modified the order of the mentioned models for better understanding, in both the abstract and Results section (14-16th lines of Abstract and 17-19th lines of Results: Study population and treatment in ICU).

* The observation periods differ between the 2 models with 7 months more in the mandatory critical care consultation model. Please explain and discuss? This resulted in a different number of patients between groups, i.e. 987 vs 539 patients.

We thank the reviewer for this valuable comment. The duration of “before” staffing model was about several months longer than the “after” staffing model and therefore, almost twice as more patients were enrolled in the mandatory consultation model compared to the closed unit model. This difference was the result of the timing of our study enrollment. We planned this study as soon as we believed we have gathered enough data to compare the different staffing models. 

We understand the reviewer’s concern and added this as a limitation of this study. “Third, there are almost twice as more patients in the mandatory consultation model compared to the closed unit model. This was due to differences in the duration of the before and after staffing models. While equal-sized groups have maximal statistical power, we believe that the smaller group of our study (n=539) was still large enough sample to detect clinical significant differences between the two groups.” (50-54th lines of Discussion). 

* How many patients were not include in these 2 periods? In other words , what was the ratio or sample size of this cohort presented/studied in comparison with the total number of patients who were admitted to ICU? 

In total, 1,076 patients in mandatory critical care consultation model group and 581 patients in closed unit model group were admitted to ICU during study period. Of them, 89 (8.3%) patients in mandatory critical care consultation model group and 42 (7.2%) patients in closed unit model group) were excluded due to incomplete medical records. The proportion of excluded patients in each group does not seem to differ significantly. We added this to the manuscript in 1-4th lines of Results: Study population and treatment in the ICU.

* What was exactly the assumption or hypothesis before starting the study?

The aim of this study was to compare closed unit model to mandatory critical care consultation model and to evaluate the superiority of the closed unit model in the outcomes of critically ill Asian patients. We have shown in a large-scale Asian population that closed unit model is associated with lower ICU mortality and shorted ICU LOS. We have added this comment in the 27-30th lines of Introduction.

* Ad Introduction, Line13 please write Pronovost instead of Pronovist

Thank you for correcting our mistake. Changes were made in 13th line of Introduction as suggested. 

* Ad Methods: L9 …during day and was responsible…? Is here missing maybe “during day and night, and was…” Do you mean 7 x 24 hours closed format/dedication, and weekend included? 

We apologize for the confusion. The intensivists of this study stayed in the ICU only during the weekday daytime, and not during the weekends and night hours. We have corrected 'during day' to 'during the weekday daytime'.

* Were there any differences regarding the time of discharge, i.e. outside time schedule 8:00-18:00 hours, with a possible association in (repeated) admission within 48 hours after discharge?

We thank the reviewer for this comment. Unfortunately, data regarding the time of discharge was not available for analysis. The rate of readmission (defined as readmission within 48 hours of ICU discharge) was not significantly different between the two groups. We added this result to the manuscript (26-27 lines of Results: Study population and treatment in the ICU and Table 2). 

* Ad statistical analysis: L7… to the five reasons… Here please describe in short what is meant? For instance: In short, ……..are the 5 organ related failures plus others?

We clarified the five reasons in the Methods section as “the reasons of ICU admission: respiratory diagnosis, cardiovascular diagnosis, acute kidney injury, sepsis, and neurologic diagnosis.” 

* Ad Results L7: patients in CUM had higher APACHE II scores. Please explain? Could it be possible that with more dedication of the intensivists present there was more time to score the APACHE better, which could have led to the this difference?

The APACHE II scores used in the study was measured by the charge nurse on duty. Therefore, differences in APACHE II scores cannot be explained by dedicated intensivists better scoring the patients. It is more likely that active admission triage of the intensivists in the closed unit model led to the admission of patients with higher severity. We added this explanation in the 21-24th lines of Discussion.

* Ad Table 1 ECOG-scale is presented as significantly different between groups, but with a mean of 3.0 vs 3.3. Statistically different, but I doubt whether of any clinical importance. Is it possible to divide between patients with ECOG≤2 and ≥3 for either group?

We thank the reviewer for this valuable comment. Proportion of patients with ECOG grade ≥3 was also significantly higher in the closed unit model group compared to the mandatory critical care consultation group (84.8% vs. 73.0%; p < 0.001) and we have added this information in the Results section lines 10-11. However, taking into account that mean ECOG grade was used in the multiple regression analysis, we ask the reviewer to consider leaving ECOG grade in Table 1 as it is. 

* Table 1 Others with a remark, with stated “etc” What is the quantity of etc and what is exactly meant by etc? In total it is >12.5%

In the medical records, the admission diagnosis of the patients admitted to the ICU was coded into 6 categories (respiratory, cardiovascular, acute kidney injury, sepsis, neurologic, and others). Others included gastrointestinal bleeding, for close observation after surgery or procedure, psychiatric, poisoning and etc. We have added this description under Table 1. We apologize for not being able to describe the ‘others’ category in detail due to the retrospective design of this study. 

* So LOS and MV duration are in favor of MCCCM. This is important for your ICU and conform previous study results. Whereas, when looking at patients with neurologic disorders (Table 5) the total number of patients may have not been enough to reach significance (p=0.074) in favor of MCCCM. Please explain and describe?

LOS and MV duration was mostly shorter in the closed unit model compared to the mandatory critical care consultation model. We agree with the reviewer that statistical significance could not be obtained in several specific admission diagnosis groups (including neurologic diagnosis) due to the lack of sample size. In total, LOS in ICU of survivors was significantly shorter in the closed unit model (5.5 days vs. 6.7 days; p = 0.005) compared to the mandatory critical care consultation model group and these results are consistent with other studies (Multz et al. Am J Respir Crit Care Med 1998;157:1468-73, Ogura et al. J Intensive Care 2018;6:57).

---

## [Decision Letter · Decision Letter 1]

10 Sep 2021

PONE-D-21-19587R1Impact of staffing model conversion from a mandatory critical care consultation model to a closed unit model in the medical intensive care unitPLOS ONE

Dear Dr. Lee,

Thank you for submitting your manuscript to PLOS ONE. After careful consideration, we feel that it has merit but does not fully meet PLOS ONE’s publication criteria as it currently stands. Therefore, we invite you to submit a revised version of the manuscript that addresses the points raised during the review process.

Please revise accordingly.

We look forward to receiving your revised manuscript.

Kind regards,

Academic Editor

PLOS ONE

Journal Requirements:

Reviewers' comments:

Reviewer's Responses to Questions

**Comments to the Author**

1. If the authors have adequately addressed your comments raised in a previous round of review and you feel that this manuscript is now acceptable for publication, you may indicate that here to bypass the “Comments to the Author” section, enter your conflict of interest statement in the “Confidential to Editor” section, and submit your "Accept" recommendation.

Reviewer #1: All comments have been addressed

Reviewer #2: All comments have been addressed

2. Is the manuscript technically sound, and do the data support the conclusions?

Reviewer #1: Yes

Reviewer #2: Yes

3. Has the statistical analysis been performed appropriately and rigorously? 

Reviewer #1: I Don't Know

Reviewer #2: Yes

4. Have the authors made all data underlying the findings in their manuscript fully available?

Reviewer #1: No

Reviewer #2: Yes

5. Is the manuscript presented in an intelligible fashion and written in standard English?

Reviewer #1: Yes

Reviewer #2: Yes

6. Review Comments to the Author

Reviewer #1: I would like to thank the authors for addressing the reviewers' comments. In my opinion, a final check for linguistic flaws (e. g. "twice as more patients" in the new paragraph of the limitations section schould be changed to "twice as many") ought to be the last step before acceptance.

Reviewer #2: Thank you for submitting your revisions and addressing all recommendations by reviewers. No further suggestions on my end.

7. PLOS authors have the option to publish the peer review history of their article (what does this mean?). If published, this will include your full peer review and any attached files.

Reviewer #1: No

Reviewer #2: No

---

## [Author Response · Author response to Decision Letter 1]

24 Sep 2021

Dear Dr. Emily Chenette,

Editor-in-Chief of PLoS One

We wish to thank the editor and reviewers for their response to our work and their constructive comments. We have addressed the concerns that were raised through further editing of the manuscript and believe it has significantly improved as a consequence of this process. We have highlighted all changes in the revised manuscript and have addressed all referees’ comments in the point-by-point responses below. We greatly appreciate the opportunity to submit this revised review manuscript and hope it is suitable for publication in PLoS One.

4. Have the authors made all data underlying the findings in their manuscript fully available?

Reviewer #1: No

We apologize that the data cannot be shared publicly due to potentially identifying or sensitive information. Requests for data access may be sent to Seoul National University Institutional Data Access / Ethics Committee (contact via Tel: 82-2-2072-0694) for researchers who meet the criteria for access to confidential data.

6. Review Comments to the Author

Reviewer #1: I would like to thank the authors for addressing the reviewers' comments. In my opinion, a final check for linguistic flaws (e. g. "twice as more patients" in the new paragraph of the limitations section schould be changed to "twice as many") ought to be the last step before acceptance.

We thank the reviewer for this valuable comment. We performed grammatical corrections on the revised text, and as a result, the following sentences were corrected.

Introduction 27-30th line: Corrected “The aim of this study was to compare closed unit model to mandatory critical care consultation model and to evaluate the superiority of the closed unit model in the outcomes of critically ill Asian patients.” to “This study aimed to compare the closed unit model to the mandatory critical care consultation model and evaluate the superiority of the closed unit model in terms of outcomes in critically ill Asian patients.”

Result 1st-2nd line: Corrected “1,076 patients in mandatory critical consultation model group and 581 patients in closed unit model group” to “1,076 and 581 patients in the mandatory critical consultation model group and the closed unit model group, respecively” 

Result 3rd-5th line: Corrected “89 [8.3%] patients in mandatory critical care consultation model group and 42 [7.2%] patients in closed unit model group” to “89 [8.3%] and 42 [7.2%] patients in mandatory critical care consultation model group and the closed unit model group, respectively” 

Discussion 21th line: Corrected “Despite patients having higher APACHE II scores at admission” to “Although the patients had higher APACHE II scores at admission”

Discussion 41-42th line: Corrected “It is impossible to completely exclude the possibility that outcome changes may have been from advances in medicine rather than changes in staffing models” to “Further, the outcome changes might have been due to advances in medicine rather than changes in staffing models”

Discussion 49-54th line: Corrected “Third, there are almost twice as more patients in the mandatory consultation model compared to the closed unit model. This was due to differences in the duration of the before and after staffing models. While equal-sized groups have maximal statistical power, we believe that the smaller group of our study (n=539) was still large enough sample to detect clinical significant differences between the two groups” to “Third, the number of patients in the mandatory consultation model group was twice as high as that than in the closed unit model group. This was due to differences in time period before and after changing the staffing model. Although equal-sized groups have maximal statistical power, we believe that the smaller size of the group in our study (n=539) was large enough to detect clinically significant differences between the two groups”.

Reviewer #2: Thank you for submitting your revisions and addressing all recommendations by reviewers. No further suggestions on my end.

Thanks for your thoughtful comment.

---

## [Decision Letter · Decision Letter 2]

13 Oct 2021

Impact of staffing model conversion from a mandatory critical care consultation model to a closed unit model in the medical intensive care unit

PONE-D-21-19587R2

Dear Dr. Lee,

We’re pleased to inform you that your manuscript has been judged scientifically suitable for publication and will be formally accepted for publication once it meets all outstanding technical requirements.

Kind regards,

Academic Editor

PLOS ONE

Additional Editor Comments (optional):

Reviewers' comments:

Reviewer's Responses to Questions

**Comments to the Author**

1. If the authors have adequately addressed your comments raised in a previous round of review and you feel that this manuscript is now acceptable for publication, you may indicate that here to bypass the “Comments to the Author” section, enter your conflict of interest statement in the “Confidential to Editor” section, and submit your "Accept" recommendation.

Reviewer #2: All comments have been addressed

Reviewer #4: All comments have been addressed

2. Is the manuscript technically sound, and do the data support the conclusions?

Reviewer #2: Yes

Reviewer #4: Yes

3. Has the statistical analysis been performed appropriately and rigorously? 

Reviewer #2: Yes

Reviewer #4: Yes

4. Have the authors made all data underlying the findings in their manuscript fully available?

Reviewer #2: Yes

Reviewer #4: Yes

5. Is the manuscript presented in an intelligible fashion and written in standard English?

Reviewer #2: Yes

Reviewer #4: Yes

6. Review Comments to the Author

Reviewer #2: Thank you for resubmitting your revisions in response to reviewers' comments. I have no additional suggestions.

Reviewer #4: Although the concept that closed ICU are better than open ICU is not novel and it has been well demonstrated in numerous studies over the past 2 decades leading to a closed ICU model at many if not most US hospitals, the authors make an interesting point that this is one of the few to study in detail the two models in Asia. This article is well written and would be of value to the literature as it validates previous findings in a south east Asia where medical training and practices may differ considerably compared to the US and other countries were the previous studies were conducted.

7. PLOS authors have the option to publish the peer review history of their article (what does this mean?). If published, this will include your full peer review and any attached files.

Reviewer #2: No

Reviewer #4: No

---

## [Editor Report · Acceptance letter]

19 Oct 2021

PONE-D-21-19587R2 

Impact of staffing model conversion from a mandatory critical care consultation model to a closed unit model in the medical intensive care unit 

Dear Dr. Lee:

I'm pleased to inform you that your manuscript has been deemed suitable for publication in PLOS ONE. Congratulations! Your manuscript is now with our production department. 

Kind regards, 

on behalf of

Dr. Robert Jeenchen Chen 

Academic Editor

PLOS ONE